# Gut Microbiome—How Does Two-Month Consumption of Fiber-Enriched Rolls Change Microbiome in Patients Suffering from MASLD?

**DOI:** 10.3390/nu16081173

**Published:** 2024-04-15

**Authors:** Karolina Kaźmierczak-Siedlecka, Dominika Maciejewska-Markiewicz, Maciej Sykulski, Agata Gruszczyńska, Julia Herman-Iżycka, Mariusz Wyleżoł, Karolina Katarzyna Petriczko, Joanna Palma, Karolina Jakubczyk, Katarzyna Janda-Milczarek, Karolina Skonieczna-Żydecka, Ewa Stachowska

**Affiliations:** 1Department of Medical Laboratory Diagnostics—Fahrenheit Biobank BBMRI.pl, Medical University of Gdansk, 80-211 Gdansk, Poland; leokadia@gumed.edu.pl; 2Department of Human Nutrition and Metabolomics, Pomeranian Medical University, 71-460 Szczecin, Poland; dominika.maciejewska.markiewicz@pum.edu.pl (D.M.-M.); karolina.jakubczyk@pum.edu.pl (K.J.); katarzyna.janda.milczarek@pum.edu.pl (K.J.-M.); 3Department of Medical Genetics, Medical University of Warsaw, 02-106 Warsaw, Poland; 4Department of Medicine, Division of Oncology, Washington University School of Medicine, St. Louis, MO 63110, USA; a.gruszczynska@genixone.eu; 5GenXone S.A., ul. Kobaltowa 6, 62-002 Złotniki, Poland; j.hermanilzycka@genixone.eu; 6Department of General, Vascular and Oncological Surgery, Medical University of Warsaw, 02-091 Warsaw, Poland; mariusz.wylezol@wum.edu.pl; 7Translational Medicine Group, Pomeranian Medical University, 70-204 Szczecin, Poland; katarzyna.kozlowska.petriczko@pum.edu.pl; 8Department of Gastroenterology and Internal Medicine, SPWSZ Hospital, 71-455 Szczecin, Poland; 9Department of Biochemical Sciences, Pomeranian Medical University in Szczecin, Broniewskiego 24, 71-460 Szczecin, Poland; joanna.palma@pum.edu.pl

**Keywords:** gut microbiota, nonalcoholic fatty liver disease, fiber, short-chain fatty acids

## Abstract

Background: The occurrence of metabolic dysfunction-associated steatotic liver disease (MASLD) is a growing global problem which commonly affects patients with co-existing diseases/conditions, such as type 2 diabetes and dyslipidemia. The effective treatment of MASLD is still limited; however, diet plays a significant role in its management. There are multiple beneficial properties of dietary fiber, including its ability to modify the gut microbiome. Therefore, the aim of this study was to determine the effect of the consumption of fiber-enriched rolls on the gut microbiome and microbial metabolites in patients suffering from MASLD. Methods: The participants were recruited according to the inclusion criteria and were required to consume fiber-enriched rolls containing either 6 g or 12 g of fiber. There were three assessment timepoints, when the anthropometric and laboratory parameters were measured, and 16s on nanopore sequencing of the fecal microbiome was conducted. Results: Firmicutes and Bacteroidetes were the most abundant phyla in the patients living with MASLD. It was demonstrated that the amount of short-chain fatty acids (SCFAs) changed after the consumption of fiber-enriched rolls; however, this was strongly associated with both the timepoint and the type of SCFAs—acetate and butyrate. Additionally, the high-fiber diet was related to the increase in phyla diversity (*p* = 0.006571). Conclusions: Overall, the introduction of an appropriate amount of fiber to the diet seems to be promising for patients suffering from MASLD due to its ability to create an improvement in gut microbiome-related aspects.

## 1. Introduction

Metabolic dysfunction-associated steatotic liver disease (MASLD)—previously referred to as NAFLD or non-alcoholic fatty liver disease—affects over one-third of adults [1,2]. The pathogenesis and development of MASLD is a complicated process which involves several factors, such as overweight, obesity, the overall presence of metabolic syndrome, and genetic factors [3,4]. Moreover, MASLD is closely related to an unfavorable composition of the intestinal microbiota, known as gut dysbiosis. For instance, the low abundance of the *Ruminococcaceae* family has been observed in patients with MASLD [5]. They had less *Coprococcus*, *Faecalibacterium*, and *Ruminococcus*, while *Prevotella*, *Escherichia*, and *Streptococcus* levels were increased. The Proteobacteria phylum (especially the *Escherichia coli* and *Enterobacteriaceae* families) overgrowth might be responsible for the increase in intestinal permeability and portal lipopolysaccharide S (LPS) levels, followed by inflammasome activation and liver injury. Additionally, *E. coli* and other *Enterobacteriaceae* families are ethanol producers, which might lead to the endogenous ethanol overproduction that is involved in the development of nonalcoholic steatohepatitis (NASH) [6]. Simultaneously, the vulnerability to the development of MASLD is mutable, and disease progression may be limited by several factors, including physical activity, intestinal microbiota products, inherited factors (i.e., genetic/epigenetic), and habitual diet [7].

Currently, there is no effective pharmacological therapy for the treatment of MASLD; therefore, lifestyle modification consisting of diet, exercise, and weight loss has been advocated to treat patients with MASLD [1,7]. The universal diet that represents the gold standard in preventive medicine is the Mediterranean diet (MD), which is characterized mainly by the consumption of plant-based foods and fish and the reduced consumption of meat and dairy products [8]. Notably, the most metabolically active components included in the MD diet are not absorbed in the small intestine. For a healthy adult on a typical Western diet, about 50–60 g of MAC reach the large intestine every day. Notably, about 20 g of this is plant (non-starchy) fiber. Around 75% of this fiber is metabolized in the large intestine. Vegetable fiber is the key MAC. Fiber, together with other components, is the main source of energy for gut microorganisms. Different types of fiber have other effects on the microbiota. For instance, fructooligosaccharides are bifidogenic and increase *Lactobacilli*. Galacto-oligosaccharides increase *Lactobacillaceae* and *Lachnospiraceae* but decrease *Ruminococcaceae*. Inulin increases *Prevotellaceae* and *Bifidobacteria*. Nevertheless, all of the above increase the diversity of the microbiota. β-glucan, pectins, and gums increase *Bifidobacteria* as well as *Lactobacilli*. Oligofructose-enriched inulin also shows a significant increase in the relative abundance of *Bifidobacterium* [9]. The regular consumption of fiber is also associated with an improvement in the function of the intestinal barrier, which is a component of the intestinal immune system, as evidenced by the reduced level of lipo-polysaccharide (LPS) [10]. A meta-analysis published in 2020 found that introducing fiber into the diet of patients with MASLD had potential benefits through effects on weight reduction, improved insulin sensitivity, and blood levels of alanine aminotransferase (ALT) and aspartate aminotransferase (AST) [11]. However, this leads to the question of whether the introduction of a diet in the short term is able to effectively alter the intestinal microbiome. The aim of the current study is to determine the effect of the consumption of fiber-enriched rolls on the gut microbiome and microbial metabolites in patients with MASLD.

## 2. Materials and Methods

### 2.1. Study Design

This trial (ClinicalTrials.gov Identifier: NCT04520724) was conducted between July 2019 and November 2019. The participants were recruited from patients of the Sonomed Medical Centre in Szczecin (Poland). After inclusion, the participants underwent three clinical assessments, at day 0 (baseline/first timepoint), after 30 days (second timepoint), and after 60 days (third timepoint). At each timepoint, specific clinical parameters and clinical material were collected, as illustrated in Figure 1.

Twenty-six eligible participants were included. We excluded participants who at enrolment or during the study were confirmed as having the following: infection with either HBV (hepatitis B virus) or HCV (hepatitis C virus) or extreme obesity (body mass index, BMI > 35 kg/m^2^); also excluded were those who reported changes in physical activity during the study, the inability to attend controlled visits, excessive consumption of alcohol (>30 g in men, and 20 g in women per day), drug addiction, or any condition that could limit the mobility of the participant.

Anthropometric assessments were performed routinely during each of the three timepoints. The measurements included height (m) and body weight (kg), and the multifrequency bioimpedance meter Tanita BC 601 (Tokyo, Japan) was used to assess body composition (fat mass in %, lean mass in %, total body water).

At the start, the individuals participating in the project had their physical activity assessed using the IPAQ test—the majority of patients demonstrated moderate physical activity. Our patients were requested not to alter their activity levels during the study [12].

Liver stiffness (using VCTE (vibration-controlled transient elastography) and the CAP (controlled attenuation parameter)) was measured with FibroScan^®^ at the baseline and at the third timepoint by a single observer (hepatologist). This procedure was carried out after ensuring that the patients had fasted for at least six hours. The selection of either the M or XL probe for these measurements was based on the distance from the skin to the liver capsule. These assessments were performed on the liver’s right lobe, utilizing the intercostal spaces while the patient was positioned on their back, with the right arm fully extended outward. The conclusive CAP and VCTE readings represented the median of 10 separate measurements, expressed in dB/m and kPa, respectively. Only those results with an interquartile range of 40% or less were deemed accurate. The classification into low-, intermediate-, and high-grade steatosis (S1, S2, S3) was determined by specific CAP cut-off values: 234 dB/m for S1, 269 dB/m for S2, and 301 dB/m for S3.

The study protocol was approved by the ethics committee of the Pomeranian Medical University (Szczecin, Poland, KB-0012/131/19) and conformed to the ethical guidelines of the 1975 Declaration of Helsinki. The volunteers provided written informed consent before the study.

The general characteristics of the study group are presented in Table 1.

### 2.2. Dietary Guidelines

At day 0 (baseline/first timepoint), the participants underwent a 30 min individual course conducted by a licensed nutritionist who addressed the dietary guidelines on the principles of the MD, which followed the recommendations of the MD Foundation [13]. To control for consistency in the dietary instructions, the dietary course followed a specific instruction pamphlet that was subsequently given to the participants. After the dietary course, specifically formulated high-fiber rolls (buns) or low-fiber rolls were introduced into the participants’ diets, and the participants were instructed to follow the study requirements. Thus, the study products were fiber-enriched rolls (i.e., the study roll contained 12 g of fiber, whereas the control roll contained 6 g of fiber), with the following ingredients. The study rolls were characterized by: a weight around 120 g, fiber content—12 g, rye flour type 2000 BIO, vital fiber (20% plantain, 80% psyllium) BIO, apple fiber BIO, ground milk thistle BIO, natural leaven from the fermentation of rye flour type 2000, and yeast. The control rolls were characterized by: a weight around 120 g, fiber content—6 g, rye flour 2000 BIO, vital fiber (20% plantain, 80% psyllium) BIO, apple fiber BIO, ground milk thistle BIO, natural leaven from the fermentation of rye flour type 2000, and yeast. The fiber contents in the rolls were determined according to the method described by AOAC [14,15]. The total insoluble and soluble dietary fiber was determined according to the enzymatic–gravimetric method using the Fibertec 1023 device (Tecator Tech., Stockholm, Sweden). The average fiber content in the rolls was 6.6 ± 0.11 g/roll; the fat was 2.38 ± 0.11 g/roll; the protein was 20.4 ± 0.47 g/roll; and the water was 63.7 ± 0.77 g/roll.

### 2.3. Fecal Microbiota Sequenced Using 16S on Nanopore

First, we present the results of the analysis of 16S on the nanopore sequencing run, where >10G bases of filtered data with a quality of >Q10 16S amplicons were obtained. The given dataset consisted of two groups: 1) low-fiber (LFIB group—6 g fiber in roll) and 2) high-fiber (HFIB group—12 g fiber in roll). Each group contained 13 patients. As was mentioned above, each patient was measured three times: before diet (T1), after one month of being on the diet (T2), and after two months of being on the diet (T3). The total fecal microbial community DNA was extracted from 78 frozen human fecal samples using a commercially available kit (Bead-Beat Micro AX Gravity kit, A&A Biotechnology, Gdynia, Poland), based on the gravity method. The human feces were subjected to enzymatic lysis (an enzyme cocktail containing lysozyme, mutanolysin, and lysostaphin) and mechanical disruption by bead beating (1 cycle: 30 s, 6 m/s) with a BeadBlaster 24 (Benchmark Scientific, Sayreville, NJ, USA). Total DNA was extracted using ion exchange columns. The extracted DNA was purified with the Agencourt AMPure^®^ XP (Beckman Coulter, Brea, CA, USA). The DNA concentration was measured with Qubit 2.0 (Invitrogen, Carlsbad, CA, USA) using a Qubit dsDNA BR Assay Kit (Thermo Fisher Scientific, Waltham, MA, USA). The hyper-variable regions (V3–V8) (~1050 bp) of the bacterial 16S rRNA gene were amplified with the following primers: two forward primers 338Fb (5′-GTCTCGTGGGCTCGGAGATGTG-TATACTCTCTATACWCCTACGGGWGGCAGCAG-3′) and 338Fa (5′-GTCTCGTGGGCTCGGA-GATGTGTATACTCTCTATGACTCCTACGGGAGGCWGCAG-3′), and 1391R primer 5′-GTCTCGTGGGCTCGGAGATGTGTATACTCTC-TATGACGGGCGGTGTGTRCA-3′. Amplification was performed with the following PCR conditions: initial denaturation at 94 °C for 30 s, 30 cycles at 94 °C for 20 s, 55 °C for 20 s, 65 °C for 90 s, and the final extension at 65 °C for 10 min. The 16S rRNA amplicons were subjected to next-generation sequencing (NGS) using nanopore sequencing technology (Oxford Nanopore Technologies, Didcot, UK). A sequencing library was created with 1D technology using a ligation sequencing kit, and the library was run on a FLO-MIN106D flow cell. Sequencing was performed on a GridION X5 sequencer from Oxford Nanopore Technologies. Extraction of DNA, next-generation sequencing, and bioinformatic analysis were performed using genXone S.A. (Złotniki, Poland).

### 2.4. Short-Chain Fatty Acids (SCFA) Analysis

The analysis was performed on a 0.5 g fecal sample, which was then homogenized in 5 mL of water for 5 min. The pH was acidified to pH = 3 with 5M HCl; further samples were centrifuged for 20 min. The obtained samples were analyzed using gas chromatography with a flame ionization detector (FID). The SCFA analysis included: acetic acid (C2: 0), propionic acid (C3: 0), isobutyric acid (C4: 0i), butyric acid (C4: 0n), isovaleric acid (C5: 0 i), valeric acid (C5: 0 n), isocaproic acid (C6: 0 i), caproic acid (C6: 0 n), and heptanoic acid (C7: 0). The analysis was performed using an Agilent Technologies 1260 System gas chromatograph (Agilent Technologies, Santa Clara, CA, USA) on a DB-FFAP column of 30 m × 0.53 mm × 0.5 µm. Hydrogen was supplied as a carrier gas at a flow rate of 14.4 mL/min. The starting temperature was 100 °C. It was held for 0.5 min and then raised to 180 °C at a rate of 8 °C/min and held for 1 min. The temperature was then increased to 200 °C at a rate of 20 °C/min and held at 200 °C for 5 min. The fatty acids were identified by comparing their retention times with the commercially available standards.

### 2.5. Bioinformatic and Statistical Analysis

#### 2.5.1. Taxonomic Assignment of 16S Reads

The assignment was performed using the UBLAST algorithm to align the reads to those of the NCBI 16S database (https://www.ncbi.nlm.nih.gov/refseq/targetedloci/16S_process/, accessed on 30 July 2020). Top scoring hits were selected for each read. In the case of multiple hits with the same score, the assignment was performed using the lowest common ancestor of the taxids in the NCBI taxonomic tree.

#### 2.5.2. Statistical Analysis

The Wilcoxon signed rank exact test was used to compare the paired data collected at the three timepoints. All the analyses were corrected for the multiple testing using Bonferroni correction (adj. *p*-value). To compare the clinical parameters between timepoints, we used ANOVA for the paired data. We also used linear regression to predict the correlation of the two parameters. Diversity was computed using the Simpson diversity index (SDI) for phyla. A Wilcoxon two-sided rank sum test with continuity correction was run (using https://search.r-project.org/R/refmans/stats/html/wilcox.test.html (accessed on 30 July 2020) from the R stats package). Phyla agreement between the timepoints was computed based on logarithm 10-based median relative abundance among the given phyla for the low- and high-fiber groups. We used a two-tailed Wilcoxon rank sum test and a binomial two-tailed test for median difference, followed by false discovery rate (FDR) control.

## 3. Results

### 3.1. Patients’ Characteristics

The patients’ characteristics are presented in Table 2. We provided the differences in the particular parameters (both anthropometric and laboratory), considering the types of intervention at the beginning (timepoint 1) and at the end of study (timepoint 3).

### 3.2. Gut Microbiota-Derived Metabolites (SCFAs)

The changes in the SCFAs between the visits when using the LFIB and HFIB diets are shown in Table 3 and Table 4. Considering the amount of acetate in the case of intervention with 12 g of fiber, there were statistically significant changes between visits 0 vs. 1 and 0 vs. 2 (*p* < 0.05), as well as 0 vs. 2 and 1 vs. 2 (*p* < 0.05), respectively. In the case of the HFIB diet (24 g), statistically significant alterations were observed between visits 0 and 1 (*p* < 0.05) and 0 and 2 (*p* < 0.05) when considering acetate, as well as between visits 1 and 2 (*p* < 0.05) when considering butyrate.

### 3.3. Microbial Diversity

Regarding the phylum level (Figure 2), the diversity increases with diet, especially the HFIBIB diet (median before diet 0.40, low-fiber diet 0.50, high-fiber diet 0.50). The increase in phyla diversity between that before the diet and that with the HFIBIB diet is significant (Wilcoxon rank sum test, *p* = 0.006571), and the increase in phyla diversity between that before the diet and that with both the LFIB and the HFIBIB diets is significant (Wilcoxon rank sum test, *p* = 0.006996).

The bar plots below (Figure 3) represent the relative abundance of the dominant bacteria at the phylum level in fecal microbiota before the diet and in the LFIBIB and HFIBIB groups. Firmicutes and Bacteroidetes are the most abundant phyla in all three groups. On average, the LFIBIB group has the highest representation of Actinobacteria and the lowest abundance of Proteobacteria.

Box plots of the relative abundances of the 12 phyla of the LFIBIB and HFIBIB groups at the 3 timepoints were plotted using the logarithmic scale (Figure 4). It can be seen that there was an increase in some phyla between timepoints 1 and 2 as well as a decrease between timepoints 2 and 3. In the case of Actinobacteria, its level was reduced between timepoints 1 and 3 in the HFIBIB diet, whereas it increased in the LFIBIB rolls.

Agreement between the timepoints was based on the logarithm 10-based median relative abundance among the given phyla for the LFIBIB and HFIB groups (Figure 5). The dashed line represents the same median values at two timepoints. The plot shows the phyla in which the median was a non-zero value for at least one timepoint. We observed that several phyla had different median relative abundance values between the timepoints for the LFIB and HFIB groups, e.g., Verrucomicrobia between timepoints 1 and 2 for the LFIB group and timepoints 2 and 3 in the reverse direction. Similar observations were made regarding Bacteroidetes. We did not observe significant changes between timepoints in either the LFIB or the HFIB groups based on the two-tailed Wilcoxon rank sum test. The sign test (binomial two-tailed test for median difference) was followed by the false discovery rate (FDR) control results, with a significant median difference for Firmicutes (timepoints 1 vs. 2 and 1 vs. 3) and Cyanobacteria (timepoints 1 vs. 3) in the LFIB and HFIB groups when analyzed together.

## 4. Discussion

MASLD is still a serious problem with limited treatment methods. Currently, it is known as the main cause of the occurrence of chronic liver disease in developed countries [16]. The role of appropriate nutrition accompanied by regular physical activity is significant in this context [17,18]. Recently, in a systematic review and meta-analysis (n = 3037) [19], the beneficial effects of calorie-restricted interventions on liver parameters, such as ALT (*p* < 0.001) and liver stiffness (*p* = 0.01), as well as hepatic steatosis (*p* < 0.001), were observed. Similarly, it was noted that the MD decreased ALT (*p* = 0.02), the fatty liver index (*p* < 0.001), and liver stiffness (*p* = 0.05) [19]. There is also a growing overview of dietary fiber and its ability to modulate the gut microbiome. According to some of the data, dietary fiber (with oligofructose) reduces the population of pathogenic bacteria and positively affects the production of beneficial metabolites [20]. Moreover, dietary fiber promotes metabolic interactions between bacteria in the gut [21]. The consumption of fiber has multiple positive effects on, among others, the maintenance of intestinal barrier integrity (which is strongly associated with SCFAs, as discussed below) [22]. In the current study, we analyzed the effects of fiber (provided in fiber-enriched rolls) on gut microbiota and microbial metabolites in relation to the aspects of MASLD.

SCFAs (acetate, proprionate, butyrate), which are produced by particular bacteria, such as *Faecalibacterium prausnitzii*, have multiple beneficial effects [23]. They improve the appropriate functioning of intestinal barrier integrity by affecting tight junctions. SCFAs are able to modulate the mucus layer and immune system by increasing the release of anti-inflammatory mediators and inhibiting the production of pro-inflammatory cytokines. In a study by Deng et al., using a mouse model (with NASH), it was shown that SCFAs significantly decreased the level of alanine aminotransferase, as well as aspartate transaminase, in the serum [24]. In particular, sodium acetate alleviated steatosis and inflammation. Zheng et al. reported that supplementation with butyrate attenuated hepatic steatosis, which was induced by a high-fat and insufficient diet in a mouse model study [25]. In the current study, we observed a statistically significant difference in the amount of SCFAs between visits in both the LF and HF diets; however, it was only related to particular visits and the type of SCFAs—acetate and butyrate. This result can be associated with, for example, the amount of fiber or the patients ‘compliance’. Interestingly, Rau et al. reported that in NAFLD patients a higher fecal level of acetate and proprionate was noted [26]. The impairment of gut barrier integrity is associated with the translocation of gut microbiota-derived metabolites/products (such as LPS) through portal blood flow to the liver [27]. As mentioned above, butyrate improves intestinal barrier integrity; consequently, it inhibits the translocation of LPS into the blood circulation and reduces the inflammation mediated by toll-like receptor 4 (TLR-4) [28]. It should be emphasized that intestinal dysbiosis in MASLD is also related to the reduction in butyrate commensal bacterial producers. Notably, there are multiple mechanisms that can connect gut microbiota-related components and the liver (as well as MASLD). Primary and secondary bile acids, dietary fiber, extracellular vesicles, bacterial DNA, lipopolysaccharide, and others seem to be significant in that context and to be aspects of the gut–liver axis as bidirectional links [29,30,31]. Beisner et al. reported that sodium butyrate and prebiotic inulin are able to induce the expression of Paneth cell α-defensins and metalloproteinase-7 and can be beneficial in terms of diet-induced obesity in MASLD [32]. The results of another trial confirmed that prebiotic supplementation improves lipid profile and liver enzymes in patients with MASLD [33].

The gut microbiome undergoes dysbiotic alteration in MASLD patients. Recently, it was reported that its profile can even be used as an early clinical ‘warning’ of the development of NAFLD [34]. The overall gut microbial signature, which is observed in patients with MASLD, is as follows: (1) the decrease in microbial diversity as well as beneficial microbes, such as *Akkermansia muciniphila* and *F. prausnitzii*; (2) the increase in *Actinobacteria*, *Anaerobacter*, *Bacteroides*, *Blautia*, *Clostridium*, *Enterobacteriaceae*, *Klebsiella pneumoniae*, and *Prevotella*; and (3) the reduction in the Firmicutes/Bacteroidetes ratio [35,36]. In the current study, it was shown that Firmicutes and Bacteroidetes were the most abundant phyla in patients with MASLD. Interestingly, in both study groups (i.e., the low-fiber and high-fiber groups), the abundance of Firmicutes was reduced, whereas the amount of Bacteroidetes was increased. Therefore, these results may be associated with the general signature of the gut microbiome, which is observed in patients with MASLD; thus, the intervention did not provide alterations. Nevertheless, in a mouse model study, it was noted that the high-fiber diet alters the ratio of Firmicutes to Bacteroidetes, positively affecting the gut microbiome [37]. Recently, in a study by Jasirwan et al., the correlation of the Firmicutes/Bacteroidetes ratio with fibrosis and steatosis was investigated [38]. This cross-sectional study included 37 patients with MASLD. Fecal samples were collected and then analyzed using 16S rRNA sequencing. It was observed that the predominant phyla were Firmicutes, Bacteroidetes, and Proteobacteria. Moreover, a significant reduction in gut microbial diversity was noted in patients with MASLD, a high level of triglycerides, and central obesity. It was also detected that the Firmicutes/Bacteroidetes ratio was correlated with both obesity and steatosis. In the current study, the relative abundance of some bacteria increased at timepoint 1, whereas it decreased between timepoints 2 and 3.

Gut microbial diversity is reduced in multiple gastrointestinal/digestive diseases related to MASLD. Regarding the phyla, in the current study it was demonstrated that bacterial diversity after introduction of the HF diet was significantly increased. This seems to be important in the context of maintaining/restoring biodiversity of the gut microbiome. This is due to the fact that the dysbiotic profile of the gut microbiome with low microbial diversity is not desirable in these patients. Chen et al. investigated the impact of 2 months of a low-carbohydrate, high-fiber diet and education on MASLD [39]. This study investigated 44 patients divided into an intervention group (low-carbohydrate, high-fiber diet, education) and a control group (with only education). It was shown that this type of diet can reduce body mass as well as body fat. Additionally, the improvement of laboratory parameters (such as liver enzymes, uric acid, blood glucose, and lipid profile, *p* < 0.05) was also observed. Recently, in 2023, it was reported in another study that dietary fiber was associated with protection against MASLD [40].

## 5. Conclusions

Gut microbiome-related aspects are altered in patients with MASLD. By considering the phylum level, we observed that Firmicutes and Bacteroidetes were the most abundant among these patients. In summary, in the current study we showed that the level of some gut microbiota-derived metabolites—SCFAs—was changed after the consumption of fiber-enriched rolls, with different results related to the particular type of SCFAs—acetate and butyrate—and between timepoints. Moreover, we demonstrated that phyla diversity was significantly increased after the introduction of a high-fiber diet. Overall, dietary fiber can beneficially affect the gut microbiome and microbial metabolites, and it can support the treatment of MASLD patients.

## Figures and Tables

**Figure 1 nutrients-16-01173-f001:**
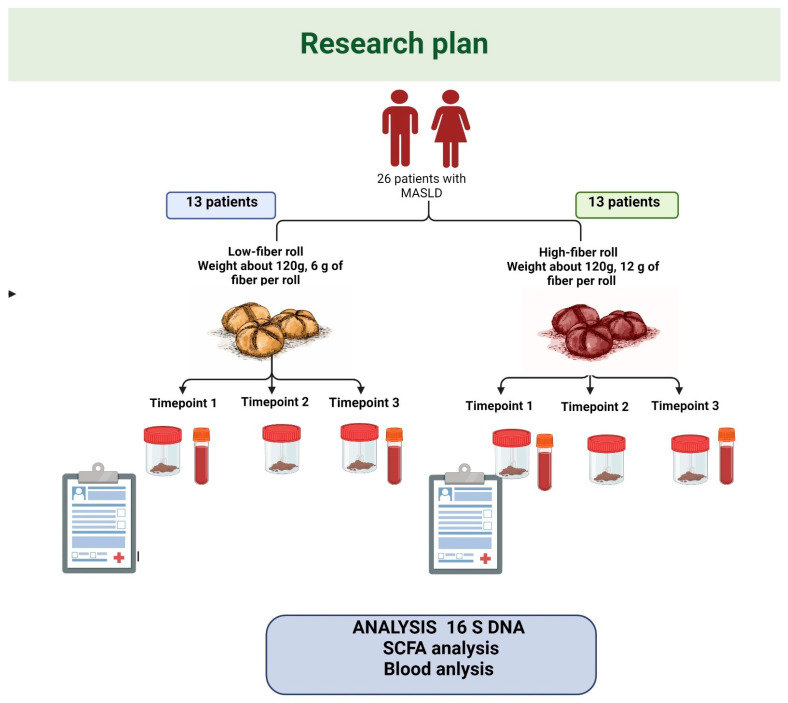
Study design, dietary guidelines, and performed analyses.

**Figure 2 nutrients-16-01173-f002:**
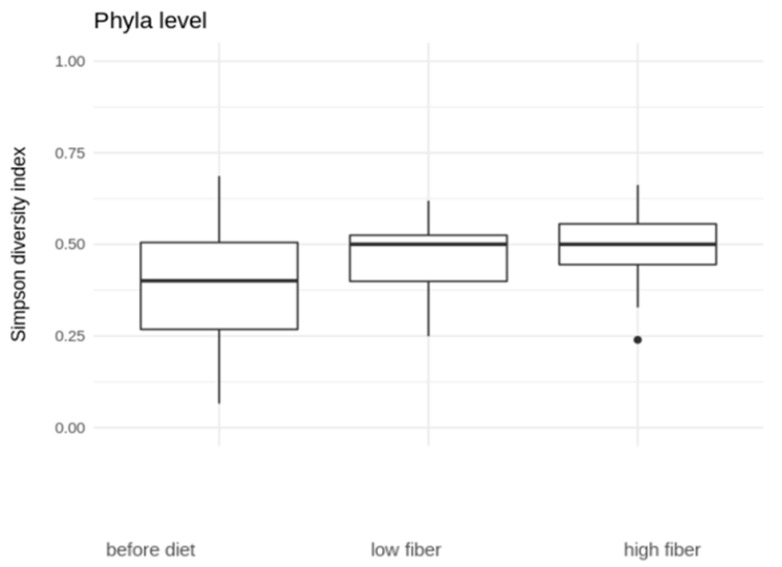
The changes in microbial diversity when considering phylum level.

**Figure 3 nutrients-16-01173-f003:**
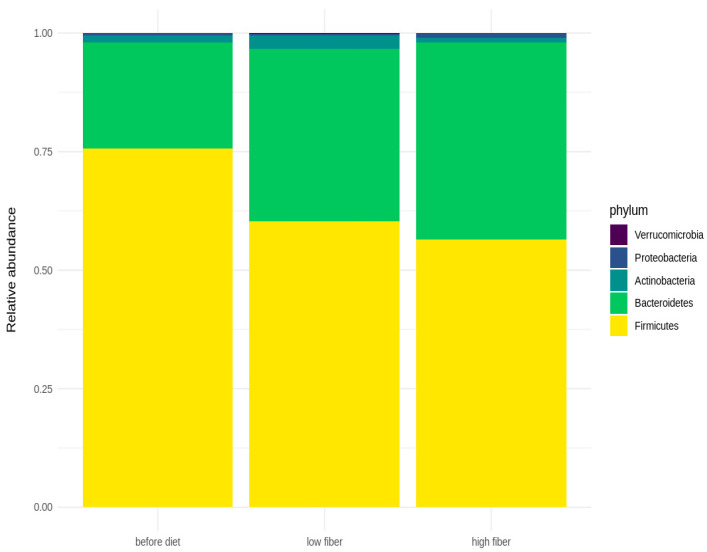
Bar plots of most abundant phyla before diet, with low-fiber diet, and with high-fiber diet.

**Figure 4 nutrients-16-01173-f004:**
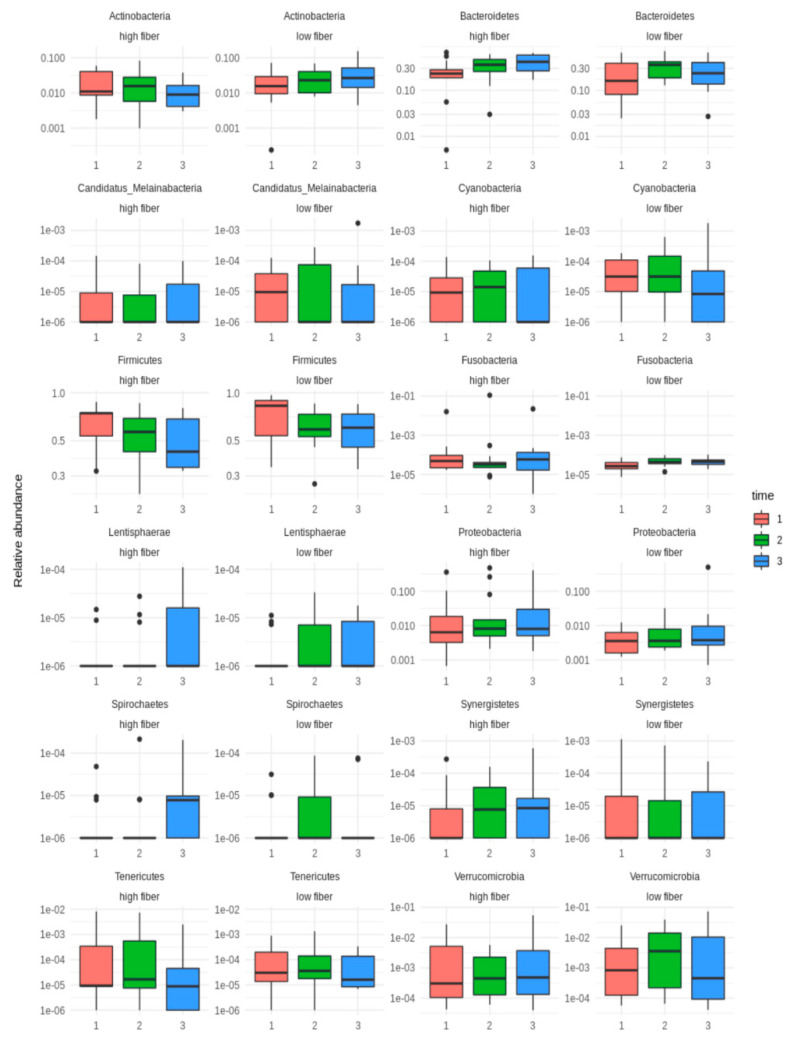
Phylum group differences between timepoints in study groups.

**Figure 5 nutrients-16-01173-f005:**
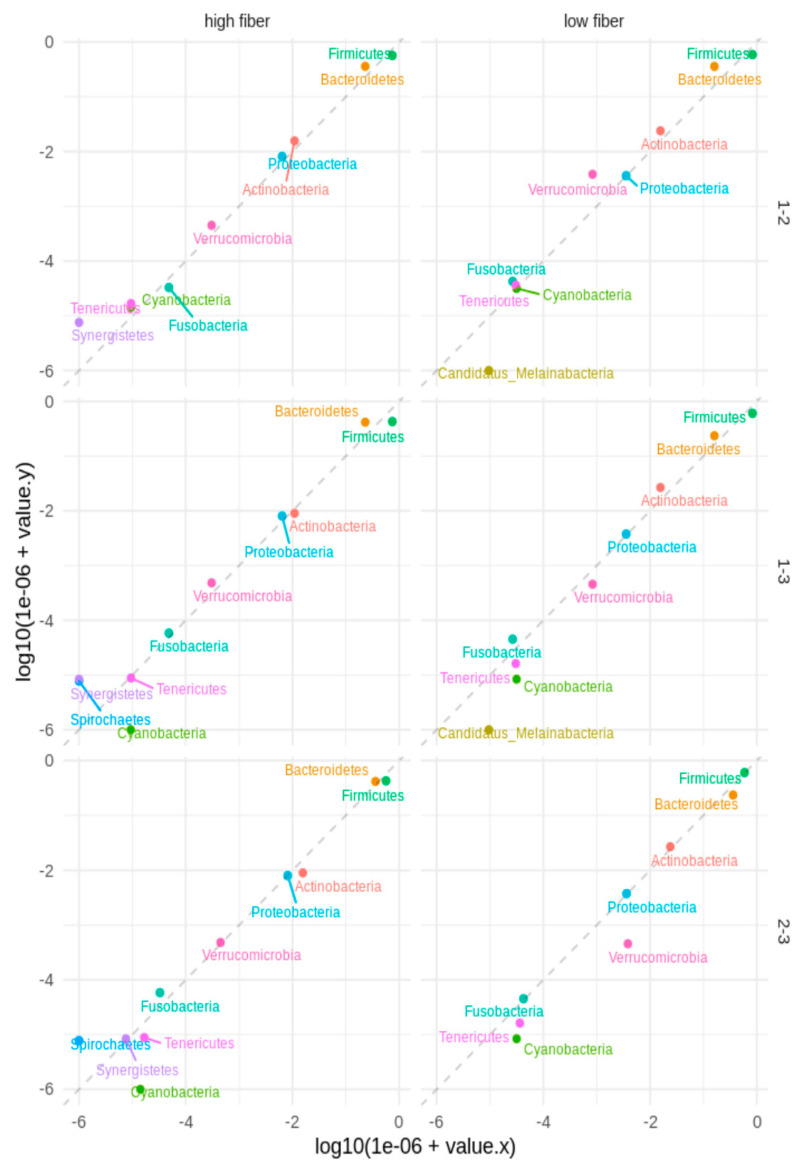
Phylum level agreement between two timepoints in each of the two study groups.

**Table 1 nutrients-16-01173-t001:** General characteristics of the study group.

Parameters	Values (Min.–Max.)
BMI	29.1 (22.2–35.7)
Body_weight	87.4 (60.3–115.6)
Body fat	29.5 (17.1–43.8)
Muscle_mass	54.6 (39.5–76.7)
Fibroscan_CAP	305.5 (242–400)
Fibroscan_elast	5.65 (3.9–9.4)
ALT	29 (11–136)
AST	22 (11–52)
GGTP	27.5 (12–70)
Total_Cholesterol	205.6 (110–394.4)
LDL	137 (43.5–282.2)
HDL	47.15 (25–71.3)

BMI—body mass index, AST—aspartate transaminase, ALT—alanine transaminase, HDL—high-density lipoprotein, LDL—low-density lipoprotein, GGTP—gamma-glutamyl-transpeptidase.

**Table 2 nutrients-16-01173-t002:** The clinical characteristics of the participants according to the types of intervention (i.e., consumption of rolls containing either 6 g of fiber or 12 g—2 times per day). **Timepoint 1 (T1) vs. timepoint 3 (T3)**.

Intervention with 12 g (LFIB) T1 vs. T3						Intervention with 24 g (HFIB) T1 vs. T3				
Parameter Intervention	Median	IQR	Median	IQR	*p*	Median	IQR	Median	IQR	*p*
Fasting glucose [mg/dL]	93	16.2	91	29.5	0.72	94.6	15.3	96.9	10.2	0.94
Total cholesterol [mg/dL]	191.4	53.2	179.9	27.1	0.18	221	59.6	197.3	47.7	**0.01**
HDL [mg/dL]	44	12.1	43.1	15.9	0.58	49.1	6	48.7	6.5	0.17
LDL [mg/dL]	125	54.5	111.9	42.1	0.21	148.9	38.9	127.7	44	0.05
TG [mg/dL]	153.3	94	129.3	92	0.11	168.2	154.2	162.3	111	0.28
ALT [U/L]	38	17	38	18	0.89	43	20	32	12	**0.04**
AST [U/L]	28	12	30	11	0.66	27	8	23	6	**0.02**
GGTP [U/L]	33	10	35	16	0.23	28	12	24	14	0.12
Fasting insulin [uU/mL]	19.1	20.3	16.1	13.9	0.66	36.8	87.2	37.6	31	0.18
Age [years]	47.5	12.3	-	-	-	47.5	14.5	-	-	-
BMI [kg/m^2^]	29.1	3.8	28.6	5.2	**0.04**	28.5	10.4	27.3	9.5	0.61

BMI—body mass index, AST—aspartate transaminase, ALT—alanine transaminase, HDL—high-density lipoprotein, LDL—low-density lipoprotein, TG—triglyceride, GGTP—gamma-glutamyl-transpeptidase; *p*—before and after intervention with LFIB and HFIB.

**Table 3 nutrients-16-01173-t003:** The changes in SCFAs between visits (i.e., 0 vs. 1, 0 vs. 2, 1 vs. 2) in case of low-fiber diet.

Intervention with 12 g (LFIB)									
SCFA mol%	C 2:0 ^A,B^			C 3:0			C 4:0 n ^B,C^		
Visit	T1	T2	T3	T1	T2	T3	T1	T2	T3
Median	64.93	68.53	62.51	19.66	17.11	18.46	14.38	12.65	18.9
IQR	14.91	10.61	14.01	4.46	3.19	3.35	7.5	9.64	7.52

A—0 vs. 1, *p* < 0.05; B—0 vs. 2, *p* < 0.05; C—1 vs. 2, *p* < 0.05.

**Table 4 nutrients-16-01173-t004:** The changes in SCFAs between visits (i.e., 0 vs. 1, 0 vs. 2, 1 vs. 2) in case of high-fiber diet.

Intervention with 24 g (HFIB)									
SCFA mol%	C 2:0 ^A,B^			C 3:0			C 4:0 n ^B*,C^		
Visit	T1	T2	T3	T1	T2	T3	T1	T2	T3
Median	60.52	63.12	56.1	20.39	19.08	20.18	14.79	12.61	15.76
IQR	9.62	10.89	9.84	4.52	4.51	2.71	4.69	7.74	7.81

A—0 vs. 1, *p* < 0.05; B—0 vs. 2, *p* < 0.05; C—1 vs. 2, *p* < 0.05; B*—0 vs. 2—*p* = 0.06.

## Data Availability

Raw data are available upon request.

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
