# Peer review of "Gut Microbiome—How Does Two-Month Consumption of Fiber-Enriched Rolls Change Microbiome in Patients Suffering from MASLD?"

_nutrients, 2024, doi:10.3390/nu16081173_

Round 1

Reviewer 1 Report

Comments and Suggestions for Authors

The manuscript is an original article that addresses the question of how two months of consumption of fiber-enriched rolls changed the microbiome in patients suffering from MASLD. The topic is interesting for researchers and clinicians, but it requires improvement by reviewing a few critical, major, and minor issues:

Critical

1. How the patients were diagnosed with MASLD is missing. In line 93, it only appears that they were "validated" by Fibroscan, a vague term. If they were not diagnosed by liver biopsy (the gold standard for MASLD), how was the diagnosis made?

2.      Explain why you did not assess the physical activity of the enrolled patients, which has a significant impact on MASLD (as it was documented in line 247, reference 18).

Both first two critical issues could invalidate the results.

3.      In Chapter 2 (Materials and Methods): The assessment of short-chain fatty acids (SCFAs) was not documented at all.

Major

1.      Lines 144-145: math does not match. Please explain more precisely why is once 40 samples, then in parentheses are 39 patients.

2.      Table 2 is unclear. When “p- for before and after intervention with LF and HF” was calculated? After 30 days, or after 60 days of intervention? The table does not show the median or IQR of the parameters before and after intervention, only one median and IQR value for LF and HF. Also, the IQR (interquartile range) is not defined as an abbreviation. Please modify the table accordingly.

3.      Chapter 4: The discussions are brief, without going into the explanation of the study results. The authors reported according to the literature, listed the results obtained, but did not explain their results.

4.      In agreement with the previous point, the discussions do not explain why certain phyla modify during the interventions. In other words, what would be the possible mechanism by which Firmicutes or Bacteroidetes were the most abundant and Firmicutes was decreased and Bacteroidetes was increased due to fiber-enriched rolls intervention?

Minor

1.      Throughout the text there are numerous hyphens within the words whose places are not appropriate (e.g. lines 98, 200 vis-its, line 111 pro-vided, line 125 in-structed, line 127-stud-ies, lines 200, 235 be-tween, line 245 occur-ing, line 250-de-creased etc…)

2.      What are the visits 0, 1, and 2 in chapter 3.2? Probably they are the time points at 0, 30, and 60 days, but it is not clear. Define exactly what they are.

3.      Lines 61-62: the abbreviations and the phrase “The most metabolically active components of the MD diet are not absorbed in the small intestine carbohydrates (MAC)” are not clear. Please reformulate the phrase.

4.      Line 74: LPS was abbreviated before in line 49. Use only abbreviations after were defined.

Comments on the Quality of English Language

Minor editing of English language required.

Author Response

Many thanks to the reviewers for their very insightful evaluation of the work. Below is our response to the reviewers 

Reviwer 1

  1. How the patients were diagnosed with MASLD is missing. In line 93, it only appears that they were "validated" by Fibroscan, a vague term. If they were not diagnosed by liver biopsy (the gold standard for MASLD), how was the diagnosis made?

Ans The answer regarding MASLD diagnostics has been added in the text and marked in red

  1. Explain why you did not assess the physical activity of the enrolled patients, which has a significant impact on MASLD (as it was documented in line 247, reference 18). Both first two critical issues could invalidate the results.

They were  assessed at the beginning of the study using the IPAQ questionnaire-our patients had moderate physical activity, and were asked not to change their activity during the study. It was added in the tekxt

  1. In Chapter 2 (Materials and Methods): The assessment of short-chain fatty acids (SCFAs) was not documented at all. – it was added

.

Major

Lines 144-145: math does not match. Please explain more precisely why is once 40 samples, then in parentheses are 39 patients. I was modified

  1.  

Table 2 is unclear. When “p- for before and after intervention with LF and HF” was calculated? I was modified

  1. After 30 days, or after 60 days of intervention? The table does not show the median or IQR of the parameters before and after intervention, only one median and IQR value for LF and HF. Also, the IQR (interquartile range) is not defined as an abbreviation. Please modify the table accordingly.

I was modified

  1. Chapter 4: The discussions are brief, without going into the explanation of the study results. The authors reported according to the literature, listed the results obtained, but did not explain their results.

We changed the disscussion

  1. In agreement with the previous point, the discussions do not explain why certain phyla modify during the interventions. In other words, what would be the possible mechanism by which Firmicutes or Bacteroidetes were the most abundant and Firmicutes was decreased and Bacteroidetes was increased due to fiber-enriched rolls intervention? I was modified

Minor

  1. Throughout the text there are numerous hyphens within the words whose places are not appropriate (e.g.lines 98, 200 vis-its, line 111 pro-vided, line 125 in-structed, line 127-stud-ies, lines 200, 235 be-tween, line 245 occur-ing, line 250-de-creased etc…) I was modified

  1. What are the visits 0, 1, and 2 in chapter 3.2? Probably they are the time points at 0, 30, and 60 days, but it is not clear. Define exactly what they are. It was changed
  2. Lines 61-62: the abbreviations and the phrase “The most metabolically active components of the MD diet are not absorbed in the small intestine carbohydrates (MAC)” are not clear. Please reformulate the phrase. I was modified

  1. Line 74: LPS was abbreviated before in line 49. Use only abbreviations after were defined. I was modified

Reviewer 2 Report

Comments and Suggestions for Authors

This manuscript explores the effects of fiber-enriched rolls on the gut microbiome and microbial metabolites in patients with Metabolic Dysfunction-Associated Steatotic Liver Disease (MASLD). Through a controlled study, participants consumed rolls with varying fiber contents, and their gut microbiomes were analyzed at multiple intervals. The study found that dietary fiber significantly impacts the abundance of certain gut bacteria and the production of short-chain fatty acids, particularly acetate and butyrate. These findings suggest that incorporating an appropriate amount of dietary fiber can enhance gut microbiome diversity and potentially offer a beneficial approach to managing MASLD, highlighting the critical role of diet in the treatment of this condition. 
 do not have any particular issues with manuscript; just a few suggestions: 
 I do not think figure 5 is the best representation of the data. It is slightly confusing; but after you know about it, it does seem superfluous. The authors might be better working with box plots with the taxa that are interesting. 
 There might not be drastic changes in the microbiome, the authors might be able to explore the functional potential using tools that can infer functions from 16S data (including genome scale models).

Author Response

Rev 2

Many thanks to the reviewer for their very insightful evaluation of the work. Below is our response to the reviewers 

This manuscript explores the effects of fiber-enriched rolls on the gut microbiome and microbial metabolites in patients with Metabolic Dysfunction-Associated Steatotic Liver Disease (MASLD). Through a controlled study, participants consumed rolls with varying fiber contents, and their gut microbiomes were analyzed at multiple intervals. The study found that dietary fiber significantly impacts the abundance of certain gut bacteria and the production of short-chain fatty acids, particularly acetate and butyrate. These findings suggest that incorporating an appropriate amount of dietary fiber can enhance gut microbiome diversity and potentially offer a beneficial approach to managing MASLD, highlighting the critical role of diet in the treatment of this condition. 
 do not have any particular issues with manuscript; just a few suggestions: 
 I do not think figure 5 is the best representation of the data. It is slightly confusing; but after you know about it, it does seem superfluous. The authors might be better working with box plots with the taxa that are interesting. 
 There might not be drastic changes in the microbiome, the authors might be able to explore the functional potential using tools that can infer functions from 16S data (including genome scale models).

I was modified

Round 2

Reviewer 1 Report

Comments and Suggestions for Authors

The authors addressed all my observations and comments, as such I agree with the publication of the article.

Comments on the Quality of English Language

Minor editing of English language is required.